# Inhibiting NADPH Oxidases to Target Vascular and Other Pathologies: An Update on Recent Experimental and Clinical Studies

**DOI:** 10.3390/biom12060823

**Published:** 2022-06-13

**Authors:** Anthony L. Sylvester, David X. Zhang, Sophia Ran, Natalya S. Zinkevich

**Affiliations:** 1Department of Biology, University of Illinois at Springfield, Springfield, IL 62703, USA; asylv2@uis.edu or; 2Department of Medicine, Cardiovascular Center, Medical College of Wisconsin, Milwaukee, WI 53226, USA; xfzhang@mcw.edu; 3Department of Microbiology, Immunology and Cell Biology, Southern Illinois University School of Medicine, Springfield, IL 62702, USA; sran@siumed.edu

**Keywords:** ROS, NADPH oxidase, NOX

## Abstract

Reactive oxygen species (ROS) can be beneficial or harmful in health and disease. While low levels of ROS serve as signaling molecules to regulate vascular tone and the growth and proliferation of endothelial cells, elevated levels of ROS contribute to numerous pathologies, such as endothelial dysfunctions, colon cancer, and fibrosis. ROS and their cellular sources have been extensively studied as potential targets for clinical intervention. Whereas various ROS sources are important for different pathologies, four NADPH oxidases (NOX1, NOX2, NOX4, and NOX5) play a prominent role in homeostasis and disease. NOX1-generated ROS have been implicated in hypertension, suggesting that inhibition of NOX1 may be a promising therapeutic approach. NOX2 and NOX4 oxidases are of specific interest due to their role in producing extra- and intracellular hydrogen peroxide (H_2_O_2_). NOX4-released hydrogen peroxide activates NOX2, which in turn stimulates the release of mitochondrial ROS resulting in ROS-induced ROS release (RIRR) signaling. Increased ROS production from NOX5 contributes to atherosclerosis. This review aims to summarize recent findings on NOX enzymes and clinical trials inhibiting NADPH oxidases to target pathologies including diabetes, idiopathic pulmonary fibrosis (IPF), and primary biliary cholangitis (PBC).

## 1. Introduction

Release of reactive oxygen species (ROS), which are unstable oxygen-containing molecules that react easily with other molecules, is central to the maintenance of vascular homeostasis. The enzymatic sources of ROS include nicotinamide adenine dinucleotide phosphate oxidases (NADPH oxidases, NOX), mitochondria, endothelial nitric oxide (NO) synthases, and xanthine oxidase (see Figure 1). ROS generation is regulated primarily by the NADPH oxidase (NOX) enzymes, which predominantly release hydrogen peroxide (H_2_O_2_) and superoxide (O_2_^−^). Since excessive superoxide production is implicated in numerous pathologies, such as vascular complications of diabetes [1], IPF [2] and PBC [3], NOX enzymes are of therapeutic interest as critical upstream inducers of H_2_O_2_ production.

NADPH oxidases were discovered in immune cells, such as neutrophils and macrophages, in the 1970s. Upon phagocytosis of pathogens, the enzymatic complex is activated and triggers O_2_^−^ production in an “oxidative burst” that acts to kill pathogens. Over time, enzymes with a similar function located in various tissues have been identified and subsequently grouped into the NOX family of enzymes [4]. The mitochondrial electron transport chain was soon demonstrated as another source of O_2_^−^ due to a “leaky” electron transport system, its O_2_^−^ scavenged by superoxide dismutase (SOD) into H_2_O_2_ [5].

However, the mediator of vascular tone remained unknown. In 1986, Gryglewski et al. demonstrated that “endothelium-derived relaxing factor (EDRF)” availability was reduced in the presence of O_2_^−^ but stabilized in the presence of SOD [6]. The following year, Palmer et al. and Ignarro et al. independently demonstrated that the EDRF factor that regulates endothelial vasodilation was nitric oxide (NO) [7,8].

In 1985, Blough and Zafiriou demonstrated the formation of peroxynitrite (ONOO^−^) in an alkaline aqueous solution following the reaction of NO and O_2_^−^ [9]. While ONOO^−^ was initially considered to simply be an intermediate in the formation of the highly reactive OH^−^, ONOO^−^ was soon demonstrated to be highly reactive with thiols [4]. The complex biochemistry of in vivo peroxynitrite reactions has been extensively studied. The targets of peroxynitrite and its reactive products range from oxyhemoglobin to methionine and tryptophan [10]. Peroxynitrite-modified proteins disrupt cellular homeostasis and, thus, contribute to multiple pathophysiological conditions. O_2_^−^ and ONOO^−^ were proposed to be drivers of atherosclerosis via lipoprotein oxidation in 1990 by White et al. [11], and ONOO^−^ was implicated in the process of aging in 2000 [12]. Ferrer-Sueta and others (2018) hypothesized that nitration and oxidation of proteins involved in self-recognition results in a sustained inflammatory response mediated by neutrophils and macrophages. Activation of these immune cells results in increased NADPH oxidase synthesis, which, in turn, perpetuates the peroxynitrite-modification of proteins [13].

These data provided the basis for antioxidant clinical trials attempting to improve health outcomes. These molecules are used in an attempt to “quench” the free electrons of ROS, such as O_2_^−^, into less reactive species. However, clinical trials using non-specific antioxidants have consistently failed to demonstrate measurable benefits. A 2020 meta-analysis of 43 antioxidant clinical trials showed that there was no association between all-cause mortality and cardiovascular disease (CVD) and antioxidant supplementation alone [14]. A 2022 study showed that an association between CVD risk and vitamins A and C, and zinc, was not statistically significant, though vitamin E may reduce CVD risk [15]. These inconclusive data are likely due to the need for low levels of ROS for signaling purposes as NOX-produced ROS play important roles, such as in cell proliferation and fighting infection. Chronic granulomatous disease is a condition in which NADPH oxidases are defective, causing neutrophils to be unable to kill pathogens. This leads to recurrent life-threatening infections from common pathogens, further demonstrating the role of ROS in health.

However, it is known that specific NOX isoforms are involved in various pathologies and pathology-driving pathways. NOX1 may drive vascular complications in diabetes, such as diabetic retinopathy [16], and perpetuate the proliferation of colon cancer [17]. NOX2 plays a role in insulin resistance [18] and is a mediator of angiotensin II-induced oxidative stress during hypertension [19]. NOX1, NOX2, and NOX4 are involved in initiating liver fibrosis [3], and NOX4 plays a role in mediating TGFβ-1 signaling which is crucial in IPF development [2]. These data provide the evidential basis for clinical trials aiming to improve health outcomes by inhibiting excess ROS production via NOX-inhibitors, and interventions such as exercise. In addition to recent key findings in ROS and NOX pathophysiology, these recent and ongoing clinical trials will be addressed in this review.

## 2. Role of NADPH Oxidases

### 2.1. NADPH Oxidase Description

Four NADPH oxidases (NOX1, NOX2, NOX4, and NOX5) are relevant to vascular homeostasis and pathology. NOX1 is widely distributed in various cell types, but expression is particularly abundant in the colonic epithelium and vascular smooth muscle cells [20]. It has been implicated in colon cancer progression [17] and vascular complications in diabetes [1]. The major ROS source in humans is NOX2 [21] that is highly expressed in phagocytes [22]. It is also the most widely expressed NOX isoform [19]. NOX2 contributes to endothelial dysfunction in vascular pathologies, such as insulin resistance in diabetes [23], but may also mediate phenotypic conversion of macrophages for tissue repair [24]. NOX4 is abundant in non-phagocytic cells, and it has been detected in vascular walls, fibroblasts, endothelial cells, and the kidney [19]. It mediates proinflammatory TGFβ-1 signaling in diseases such as IPF [19], but is also necessary for the polarization of macrophages [25]. NOX5 expression has been detected in lymphatic tissue, the testis, and blood vessels in humans [26]. Some studies showed that NOX5 contributes to vascular and kidney pathologies [26], but a recent study demonstrated a potential protective role of NOX5 against atherosclerosis in rabbits [27].

### 2.2. ROS and NADPH Oxidase in Recent Experimental Studies

Recent experimental data further highlight the role of NADPH oxidases and ROS in health and disease. Clinical intervention in illnesses of vascular dysfunction is limited by phenomena such as neointimal hyperplasia which drive further vascular damage. Palmitoylethanolamine (PEA) is a well- known anti-inflammatory agent and rutin (RUT) has antioxidant and vasoprotective properties. In a recent study, significant structural change in vessel morphology was observed following two weeks of carotid ligation, including ROS production and inflammatory cell infiltration [28]. Samples treated with a 1:1 ratio of PEA/RUT exhibited reduced change in vascular morphology, indicating that PEA/RUT administration was effective in attenuating inflammation, oxidative stress, vascular damage and vascular remodeling.

Thompson et al. demonstrated that microvascular function was restored in diet-induced obese mice following deletion of NOX1, independent of metabolic function [1], suggesting inhibition of NOX1 may be effective in preventing diabetic vascular complications. NOX1 is also highly expressed in colon cancer cells and supports their proliferation [17]. The 80–90% knockdown of NOX1 expression using shRNA increased tumor cell doubling time two- to three-fold without increasing apoptosis in HT-29 human colon cancer cells. A decline in hypoxia-inducible factor 1α (HIF-1α) downstream of attenuated NOX1 expression was associated with downregulation of mediators of cell proliferation and angiogenesis, including VEGF, c-MYC and c-MYB; the latter two are known oncogenes [29,30]. These data suggest that the proliferative phenotype in some colon cancers is supported by NOX1, and that NOX1 inhibition may be a therapeutic target for some colon cancers.

NADPH oxidases facilitate normal physiological processes by acting as signaling molecules. Recent studies have demonstrated their role in promoting macrophage polarization and subsequent tissue repair. The phenotypic conversion of pro-inflammatory Ly6ChiCX3CR1lo monocytes/macrophages to pro-resolving Ly6CloCX3CR1hi macrophages for liver repair is promoted by neutrophil signaling [24]. ROS, which are released by neutrophils via NOX2, are important mediators in this phenotypic conversion. Additionally, conversion was prevented in NOX2 deficient mice, indicating their role in tissue repair [22]. NOX4 has also recently been shown to be expressed in macrophages and to control their polarization in an NFκB-dependent manner [25]. NOX4 deficiency reduced the wound-healing M(IL4+IL13) macrophage population while forcing the polarization of proinflammatory M(LPS+IFNγ) macrophages. Decreased expression of NOX4 reduces STAT6 activation and promotes NFκB activity, resulting in increased NOX2 expression and associated O_2_^−^ production. Thus, NOX4 also has an anti-inflammatory role in macrophages [22].

Previous work demonstrated that NOX4-derived H_2_O_2_ contributes to endothelium-dependent-vasodilation in mouse mesenteric arteries and rat intrarenal arteries [31,32] and that bradykinin (BK)-induced dilation in human arterioles depends on Nox-derived H_2_O_2_ [33]. A recent study demonstrated that NOX4 regulates the activity of TRPV4 endothelium-dependent dilation in human adipose and coronary arterioles via phosphorylation of Ser824 [34]. GKT137831, a NOX1/NOX4 inhibitor, reduced Ach-induced dilation as did the TRPV4 inhibitor HC067047, but GKT137831 did not further reduce vasodilation following application of HC067047, indicating a similar signaling pathway. These data support the authors’ novel hypothesis that NOX4-generated H_2_O_2_ stimulates phosphorylation, activating TRPV4 to cause Ca^2+^ influx and subsequent relaxation of the endothelium by factors such as NO.

NOX5 has been largely unstudied due to its absence in rodents. However, it is known to have functions in both homeostasis and pathogenesis. NOX5 contributes to coronary artery smooth muscle cell contraction, angiogenesis and atherosclerosis progression [35,36]. A recent study demonstrated a novel role for NOX5 in protecting against the progression of atherosclerosis [27]. NOX5-deficient young male rabbits consuming an atherogenic, high-fat feed developed significantly more plaques in the thoracic aorta compared to wild-type controls. These findings are in contradiction to previous work demonstrating that NOX5 drives atherosclerosis progression, highlighting its ill-studied nature. The mechanistic details of NOX5′s potentially protective role are unknown at this time and warrant further investigation.

### 2.3. NADPH Oxidases and ROS in Disease

NADPH oxidases have been implicated in numerous pathologies and pathology-driving pathways. Pulmonary epithelial cells express NOX isoforms including NOX1, NOX2, and NOX4 [37]. IPF is characterized by increased levels of mitochondrial and NADPH oxidase ROS. The condition is exacerbated by dysfunctional mitochondria which produce excess O_2_^−^ and H_2_O_2_, increasing expression of NOX4 and TGFβ-1 signaling [2,38]. TGFβ-1 signaling promotes proinflammatory damage and collagen accumulation in the lungs [39], which in turn drives apoptosis and the formation of fibrotic tissue in IPF [40]. NOX4 additionally mediates the activity of TGFβ-1-induced cell differentiation, cardiac differentiation and transcriptional regulation [19]. Hepatocytes also express NOX isoforms, including NOX1, NOX2 and NOX4, which have been implicated in critical steps in initiating liver fibrosis, including hepatic stellate cell activation and hepatocyte apoptosis [3]. TGFβ-1 has also been demonstrated to contribute to the progression of PBC by enhancing fibrogenesis [41]. Together, these data suggest NADPH oxidases, particularly NOX4, may play a role in the progression of fibrotic pathologies, such as IPF and PBC, implicating selective NADPH oxidase inhibitors as promising therapeutic agents.

Gastroesophageal reflux disease (GERD) is a chronic digestive disease in which refluxate fluid frequently flows into the lower esophagus, causing esophagitis. GERD is the strongest known risk factor for esophageal adenocarcinoma [42]. Additionally, GERD has been implicated in the pathogenesis of IPF due to its high prevalence in IPF patients and the discovery of gastric acid components in bronchoalveolar lavage fluid of IPF patients [39,43]. Recent evidence suggests GERD pathogenesis is driven by an inflammatory environment constituted by increased production of cytokines, chemokines, ROS and a disturbed endogenous antioxidant defense system [39]. ROS generated by mitochondria and NOX1 and NOX2 have been demonstrated to contribute to the genotoxic effects of acidic bile reflux (BA/A). Elevated ROS production and associated DNA damage were attenuated with apocynin, NADPH oxidase-inhibiting gp91ds peptide, and siRNA targeting NOX1 and NOX2 [42]. These data suggest NOX1 and NOX2 inhibitors may be among promising treatments for esophageal adenocarcinomas.

The hallmarks of type 2 diabetes include hyperglycemia and insulin resistance. Insulin resistance is known to impair endothelium vasodilation and thus contributes to endothelial dysfunction [44]. ROS formation can be directly increased in tissues with hyperglycemia. Advanced glycation end-products that form in hyperglycemic tissues enhance the production of mitochondrial ROS and stimulate NADPH oxidases [23]. NADPH oxidases are further stimulated by the activation of protein kinase C and increased diacylglycerol synthesis in hyperglycemia [45]. NOX1 contributes to diabetic retinopathy [16], and NOX2 function has been demonstrated to be crucial in insulin resistance. The deletion of NOX2 in mice fed a high-fat diet significantly reduced insulin resistance [18], and O_2_^−^ production in pulmonary endothelium in mutant insulin resistant mice was significantly reduced in NOX2-deficient subjects [44]. These data demonstrate the vital role of NOX2-produced ROS in insulin resistance, and further support the role of NADPH oxidases in driving endothelial dysfunction observed in type 2 diabetes.

Oxidative stress and endothelial dysfunction are additionally driven by angiotensin II (Ang II) signaling. Ang II contributes to atherosclerosis and, in hypertension, a major risk factor for cardiovascular and cerebrovascular disease, mediates functional and structural changes in vasculature [46,47]. NADPH oxidases are activated by Ang II, playing a role in the increase in ROS in vascular cells induced by Ang II. NOX2, in particular, may be a prominent mediator of Ang II-induced oxidative stress and its harmful effects in cerebral circulation during hypertension [19].

Regulated cell death plays an important role in the pathogenesis of conditions such as neurodegenerative diseases and cancer. Various regulated cell death pathways (ferroptosis, pyroptosis, necroptosis, alkilaptosis) may share common signals, such as redox signaling. Ferroptosis is a regulated necrosis resulting in the damage and rupture of the cell membrane driven by the accumulation of iron and subsequent lipid peroxidation by lipoxygenase (ALOX) or cytochrome P450 (POR). ALOX- and POR-mediated lipid peroxidation is promoted by NADPH oxidase, particularly NOX1, NOX2, and NOX4 and mitochondrial ROS [48]. Thus, NADPH oxidase inhibition for the purpose of attenuating regulated cell death pathways involved in cancer and neurodegenerative pathogenesis warrants further investigation.

### 2.4. NADPH Oxidase-Produced H_2_O_2_ Mediates Vascular Tone in Healthy Subjects during Exercise

A novel transition from an NO- to ROS-mediated mechanism of vasodilation has been described in human microcirculation. Nitric oxide synthases (NOS) produce NO, the regulator of vascular tone in healthy adults. During exercise, a switch to NADPH oxidase-produced H_2_O_2_ to mediate vasodilation occurs [49]. This is transient in healthy subjects but permanent in patients with coronary artery disease (CAD), a condition in which coronary arteries send insufficient blood to the heart, typically due to atherosclerosis and inflammation. The switch to H_2_O_2_-mediated vasodilation, in turn, perpetuates the chronic inflammatory condition in CAD. The mechanisms responsible for the transition in endothelial factors are not fully understood. NOS produces NO in healthy adults through the catalysis of L-arginine, O_2_ and NADPH-derived electrons. Tetrahydrobiopterin (BH4) is necessary for electron transfer during this process, and, if limited, NOS will instead catalyze the formation of O_2_^−^. Superoxide reduces the availability of BH4, thus perpetuating the loop of O_2_^−^ production [23]. The formation of ONOO^−^ from O_2_^−^ reacting with NO further reduces the presence of NO and can both directly and indirectly drive oxidative stress [4]. This self-perpetuating cycle may, in turn, create a reliance on NADPH oxidase-produced H_2_O_2_ to preserve vascular tone [50].

### 2.5. RIRR, Endothelial Dysfunction and Angiogenesis

Endothelial dysfunction can also originate from several self-perpetuating ROS-producing loops. The formation of H_2_O_2_ and O_2_^−^ is primarily modulated by NADPH oxidases, the mitochondrial electron transport chain, and xanthine oxidase [23]. NOX4-produced H_2_O_2_ stimulates NOX2 to produce O_2_^−^, which in turn stimulates mitochondria to produce O_2_^−^ and H_2_O_2_ via the electron transport chain, a process termed ROS-induced ROS release (RIRR) [51]. ROS produced by mitochondria then activate NADPH oxidases [51] perpetuating this loop. ROS are also generated when hypoxanthine is oxidized to xanthine by xanthine oxidase, which is then converted to uric acid, forming H_2_O_2_ and O_2_^−^ by transferring electrons to O_2_ [23]. These mechanisms ultimately perpetuate the inflammatory environment.

ROS-induced ROS release orchestrated by NOX4, NOX2, and mitochondria converts endothelial cells (EC) from a quiescent to an angiogenic phenotype by enhancing ROS-dependent VEGF signaling [52]. Either NOX4 or NOX2 knockdown or overexpression of mito-catalase (a scavenger of mitochondria-derived H_2_O_2_) inhibited EC migration and proliferation, thus providing evidence of a feed-forward mechanism responsible for driving the angiogenic process [52]. Elucidation of this signaling pathway would enable the development of promising therapeutic strategies aiming to modulate angiogenesis to alleviate various pathologies.

Peripheral arterial disease (PAD) is a circulatory condition in which occlusion of arteries supplying lower extremities depletes blood supply and may lead to amputation. Enhancing ROS-dependent VEGF-signaling may be used as an important therapeutic approach to promote neovascularization and tissue repair in PAD patients. On the other hand, inhibiting this signaling pathway may prevent excessive angiogenesis contributing to cancer, diabetic retinopathy, and atherosclerosis [53].

## 3. Overview of Clinical Trials

### 3.1. Clinical Trials Targeting NADPH Oxidases to Improve Patient Outcomes

The following studies target NADPH oxidases to attenuate ROS production and improve patient outcomes. The goal of these interventions is not to fully eliminate the production of ROS, but rather to ensure proper levels, so that they may fulfill their physiological role as signaling molecules. Approaches may include direct interventions, such as those aiming to use NOX1/NOX4 inhibitors to slow the progression of fibrosis [54,55] or those attempting to indirectly improve markers of NADPH oxidase-mediated endothelial dysfunction [56]. Recent clinical trials inhibiting ROS production by direct administration of NADPH oxidase inhibitors, and those that utilize other methods to reduce NADPH oxidase-mediated ROS production, will be addressed in this section. The latter include clinical trials which measure endothelial function, ROS production and associated health outcomes following an exercise or surgical intervention.

### 3.2. A Trial of Setanaxib in Patients with Primary Biliary Cholangitis (PBC) and Liver Stiffness

PBC is a progressive autoimmune disorder that affects the bile ducts. Chronic inflammation of the liver in patients with PBC leads to degradation of the bile ducts and scarring of the liver tissue. There is currently no known cure for PBC. A placebo-controlled double-blind clinical trial is currently recruiting a projected 318 patients with PBC and will include an initial 52-week period in which Setanaxib, an experimental inhibitor of NOX1 and NOX4, will be administered, followed by a 52-week extension period. The trial was due to begin in December 2021 and the estimated primary completion date was 16 September 2024. Due to the role of NOX4 in RIRR, Setanaxib may reduce ROS production by inhibiting NOX4, thereby reducing inflammation in the liver. Chronic liver inflammation drives bile duct degradation and scar tissue formation; therefore, Setanaxib administration may arrest progression of PBC. The primary outcome measure of the clinical trial is a “biochemical response” to Setanaxib in a portion of the patients. A reduction in alkaline phosphatase (ALP) level to <1.67× of the upper limit of normal (ULN) and a ≥15% ALP level reduction from baseline, and total bilirubin ≤1 × ULN, would constitute a “biochemical response”. Secondary outcome measures include changes from baseline in liver stiffness, fatigue, and pruritus, as well as in any treatment-emergent adverse events (TEAEs) or adverse events of special interest (AESIs). Each of the three study arms will adhere to one of the following regimens in the initial 52-week period: 1200 mg/day Setanaxib, 1600 mg/day Setanaxib, or placebo. An interim analysis will determine if the experimental groups will have their dosage increased or decreased and if the placebo group will switch from placebo to Setanaxib in the extension period.

### 3.3. GKT137831 in IPF Patients with Idiopathic Pulmonary Fibrosis (GKT137831)

IPF is an inexorable condition in which scar tissue develops in the lungs. ROS generated by NOX enzymes are thought to be an important factor in the progression of IPF [40], and, thus, it is hypothesized that a NOX inhibitor will decrease pulmonary injury. GKT137831 is an inhibitor of the NOX1 and NOX4 enzymes. Inhibition of NOX4 by GKT137831 may potentially reduce NADPH oxidase-generated ROS by preventing the RIRR prompted by NOX4. A placebo-controlled, double-blind, randomized clinical trial using GKT137831 to treat 60 participants with IPF began on 7 September 2020. The estimated primary completion date is 31 July 2023. The primary outcome measure of this study is the level of oxidative stress measured by concentrations of o,o’-dityrosine in blood plasma relative to baseline. Secondary outcome measures include products of collagen degradation, pulmonary function (by spirometry), ambulatory ability, and any adverse events relative to baseline. Two study arms will adhere to either of the following regimens over a 24-week period: 800 mg/day GKT137831 or placebo.

### 3.4. Microvascular Dysfunction in Obesity

Impaired endothelial function is a well-established hallmark of obesity-related diseased states, such as CAD, atherosclerosis and diabetes. The principal investigator of this clinical trial previously demonstrated that: (1) the skeletal muscle of obese individuals has elevated levels of in vivo ROS compared to that of lean or overweight individuals, (2) in obese individuals, elevated in vivo ROS levels were normalized and local microvascular endothelial dysfunction was reversed by the perfusion of apocynin, an NADPH oxidase inhibitor, and (3) in vivo H_2_O_2_ production and microvascular endothelial dysfunction were ameliorated in obese individuals by aerobic exercise [57]. The two purposes of this study are to determine the effects of mitochondrial ROS and specific NOX isoforms on NADPH oxidase-dependent endothelial dysfunction, and the mechanism by which aerobic interval training reduces ROS and improves endothelial function. The investigator’s central hypothesis is that downregulation of mitochondrial ROS and NOX4 will improve endothelial function in obese individuals [56]. This study began on 20 November 2019 and the projected primary completion date is 31 July 2022. The anticipated total enrollment is 25 subjects. Microdialysis analysis of in vivo H_2_O_2_ production will be performed both before and after an 8-week aerobic interval training regimen for a total of 50 microdialysis experiments. Subjects are to be recruited, screened, enrolled and tested on an ongoing basis throughout the study period.

### 3.5. Endothelial Dysfunction and Oxidative Stress in Children with Sleep Disordered Breathing

Obstruction of the upper airway during sleep causes sleep disordered breathing (SDB) which is prevalent in both adults and children. Obstructive sleep apnoea (OSA) increases risk for CAD, hypertension and stroke in adults and SDB may significantly impact cardiovascular function in children. The principal investigator of this clinical trial previously demonstrated that NADPH oxidase activation is directly related to arterial endothelial dysfunction and that OSA-related NADPH oxidase-generated oxidative stress can be partially reversed by nasal continuous positive airway pressure treatment (CPAP). CPAP is the primary intervention for adults with OSA; however, SDB in children is typically caused by the presence of enlarged tonsils and adenoids, and thus the primary treatment available is adenotonsillectomy (AT). The principal investigator’s primary objectives were to evaluate the role of NOX activity, oxidative stress, inflammation, and endothelial function in children with SDB and how they are affected by adenotonsillectomy. This study began in February 2012 and one arm included 15 children with SDB who received AT. Flow-mediated dilation (FMD) was used to assess levels of serum 8-iso-PGF2α and sNOX2-dp before and after one month post-AT. Serum 8-iso-PGF2α and sNOX2-dp were significantly higher and FMD was lower in SDB children compared to 67 healthy controls [58]. FMD improvement and a reduction in oxidative stress in SDB children following AT supported the researcher’s hypothesis that NOX2-derived ROS production plays a role in arterial and endothelial dysfunction in SDB children. The study concluded in May 2014 and the results described in this section were published in May 2015.

### 3.6. Oral GKT137831 in Patients with Type 2 Diabetes and Albuminuria

Type 2 diabetes is a progressive disease in which hyperglycemia is strongly associated with stroke, myocardial infarction, and mortality. Hallmarks include loss of β-cell function and exacerbation of insulin resistance [59]. Diabetes is the leading cause of chronic kidney disease, frequently resulting in albuminuria [60]. Renin-angiotensin-aldosterone inhibition, through either an ACE inhibitor or angiotensin receptor antagonist, is the current treatment guideline for albuminuria, and is reported to slow the progression of kidney disease [61]. Oxidative stress due to ROS may initiate the progression of vascular and endothelial dysfunction associated with type 2 diabetes [23]. Thus, a clinical trial evaluating the efficacy of oral GKT137831, a NOX1/NOX4 inhibitor, in 200 type 2 diabetes patients with maximal inhibition of the renin-angiotensin-aldosterone system and residual albuminuria was conducted. The study began in October 2013 and concluded in March 2015. The primary outcome measures of the study were albuminuria and urine albumin-to-creatinine ratio (ACR), a useful predictor for cardiovascular outcomes and mortality in patients with diabetes [62]. Secondary outcome measures were glucose metabolism, including changes in HOMA-B, HOMA-IR, and HbA1c from baseline. Additional outcome measures included erectile dysfunction and neuropathic pain in patients with these complications. Two study arms adhered to either of the following regimens: 1100 mg capsule twice per day for six weeks followed by 2100 mg capsules twice per day for six weeks, or one capsule of placebo twice per day for 12 weeks [63]. Although this study has been completed, the results were not available to include in this publication. The authors of this study have been contacted and the results will be updated.

## 4. Discussion

Although outcomes from many of the clinical trials discussed have not yet become available, inhibition of specific Nox isoforms represent a promising line of study and warrants further investigation. As illustrated in Figure 2, the full elimination of ROS is not the ultimate goal of the treatment, as basal levels of ROS play a role as signaling molecules, allowing NADPH oxidases to communicate with other enzymes, such as NOS and mitochondria. The maintenance of vascular tone in health is mediated primarily by NO and characterized by basal ROS levels, and its transition to mediation by H_2_O_2_ is implicated in CAD. Increased NADPH oxidase activity drives pathology and reliance on H_2_O_2_ for vascular tone maintenance through excessive production of O_2_^−^ and H_2_O_2_, and, thus, NOX enzymes are promising targets for improving vascular health.

We have discussed recent clinical trials, both completed and ongoing (Table 1), in this review. These studies were selected for their demonstration of NADPH oxidase inhibition and attenuation as a promising treatment for pathologies such as diabetes and endothelial dysfunction. NOX1/NOX4 inhibitors, such as GKT137831, have demonstrated excellent tolerability and attenuation of inflammatory markers in clinical trials [65]. Demonstration that FMD was improved, and oxidative stress was reduced, in SDB children following AT, additionally support NOX2′s role in endothelial dysfunction and its inhibition as a promising therapeutic approach [58]. In vivo H_2_O_2_ reduction and improvement of endothelial dysfunction following perfusion of apocynin, or aerobic exercise, further support this approach [57].

Clinical trials studying NADPH oxidase inhibition can have several crucial limitations. First, it is difficult to obtain accurate measurements of ROS in vivo. Basal levels of ROS are very low and thus are often difficult to measure. Furthermore, techniques employed to measure ROS in vivo require high levels of skill and precision. These techniques are also quite invasive, requiring either tissue biopsies or perfusion of multiple solutions simultaneously. This naturally lends itself to low levels of adherence from subjects participating in clinical trials. Second, the low number of selective inhibitors for NOX isoforms also limits their application in clinical trials [66]. Non-selective inhibitors, such as apocynin, have been traditionally used but produce undesirable off-target effects and thus limit study of specific NOX isoform inhibition [67]. Finally, the number of completed clinical trials of NADPH oxidase inhibitors is still very limited with many currently ongoing. Thus, additional trials studying the effect of specific NOX isoform inhibitors on both healthy and CAD patients must be performed. Further screening of potential NADPH oxidase inhibitors that target specific NOX isoforms will significantly expand the arsenal of inhibitors available for future studies.

## Figures and Tables

**Figure 1 biomolecules-12-00823-f001:**
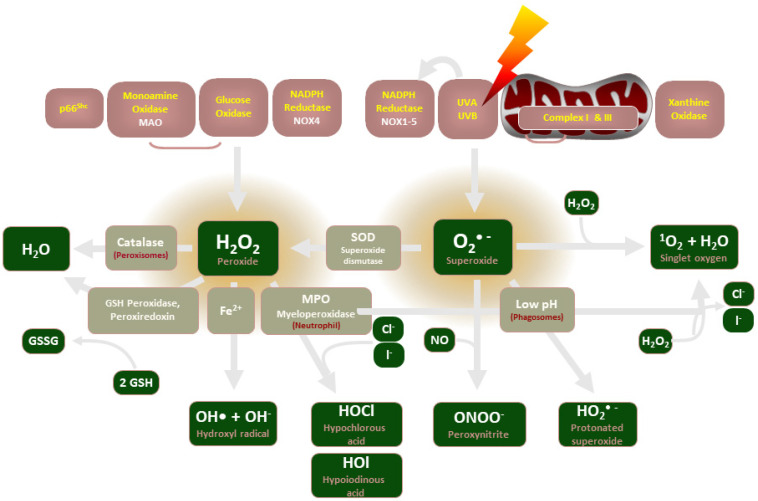
An illustration of the enzymes involved in ROS production and their common end-products.

**Figure 2 biomolecules-12-00823-f002:**
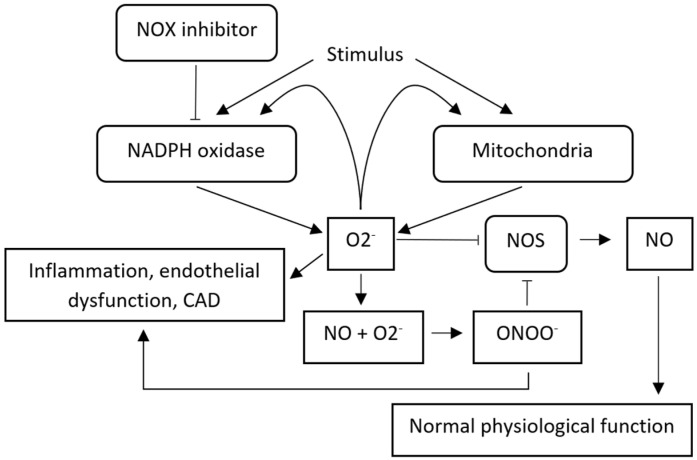
Sequence of events illustrating interactions between NADPH oxidases, mitochondria, (endothelial) NOS and their products in vascular health and disease. While excess ROS produced from NADPH oxidases contributes to pathology, note that increased activation of some NOX isoforms, such as NOX4, may serve a cardioprotective function [64]. Examples of NOX inhibitors include GKT137831, Setanaxib, and apocynin.

**Table 1 biomolecules-12-00823-t001:** Recent clinical trials targeting NADPH oxidases to improve vascular function.

Intervention	Setanaxib—A NOX1 and NOX4 Inhibitor	GKT137831—A NOX1 and NOX4 Inhibitor	Aerobic Exercise	Adenotonsillectomy (AT)	GKT137831—A NOX1 and NOX4 Inhibitor
**Regimen**	1200–1600 mg daily for 52 weeks	400 mg twice daily orally for 24 weeks	Intervals of 70–90% max HR for 30 min 3 times a week for 8 weeks	-	100 mg twice daily for first 6 weeks of treatment, and 200 mg twice daily for next 6 weeks
**Target**	Reduce liver inflammation and bile duct injury in patients with primary biliary cholangitis (PBC)	Reduce pulmonary injury in patients with idiopathic pulmonary fibrosis (IPF)	Reduce NOX isoform expression and mitochondrial ROS thereby improving endothelial function	Reverse adverse cardiovascular effects of obstructive sleep apnea and NADPH oxidase-associated endothelial function	Evaluate efficacy of oral GKT137831 in Type-II diabetic patients with residual albuminuria
**Status**	Recruiting	Ongoing	Ongoing	Completed	Completed
**(Estimated) start date**	1-December-2021	7-September-2020	20-November-2019	12-February-2012	2013
**(Estimated) completion date**	16-September-2024	31-July-2023	31-July-2022	1-May-2014	1-March-2015
**No. of patients**	318	60	25	15	200
**Results**	-	-	-	See Section 3.5	See Section 3.6

## Data Availability

No new data were created or analyzed in this study. Data sharing is not applicable to this article.

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
