# Peer review of "Inhibiting NADPH Oxidases to Target Vascular and Other Pathologies: An Update on Recent Experimental and Clinical Studies"

_biomolecules, 2022, doi:10.3390/biom12060823_

Round 1
Reviewer 1 Report
This review does not report any relevant information. Only general statements are reported (e.g.ONOO- was implicated in various pathologies as well as the process of aging in 2000) without describing anything in detail. Only 24 references are present, despite the fact that NADPH oxidases have been studied in thousands of studies. Therefore this paper must be rejected.
Author Response
This review does not report any relevant information. Only general statements are reported (e.g.ONOO- was implicated in various pathologies as well as the process of aging in 2000) without describing anything in detail. Only 24 references are present, despite the fact that NADPH oxidases have been studied in thousands of studies. Therefore this paper must be rejected.
We would like to thank the reviewer for the feedback and helpful critique. The manuscript has been updated to include more specific information and remove generalized statements and conclusions. The review now contains sections describing ROS and NADPH oxidases in experimental studies, specific human diseases, and disease-relevant pathways. This information is then used to explain why NADPH oxidase inhibition has a potential to improve patient outcomes in the clinical trials we described in the manuscript. Additionally, the references in the revised review have been significantly expanded with more recent references added to reflect numerous studies on NADPH oxidases. The total number of references has been more than doubled.
“The complex biochemistry of in vivo peroxynitrite reactions has been extensively studied. The targets of peroxynitrite and its reactive products range from oxyhemoglobin to methionine and tryptophan [10]. Peroxynitrite-modified proteins disrupt cellular homeostasis and thus contribute to multiple pathophysiological conditions. O2- and ONOO- were proposed to be drivers of atherosclerosis via lipoprotein oxidation in 1990 by White et al. [11], and ONOO- was implicated in the process of aging in 2000 [12]. Ferrer-Sueta and others (2018) hypothesized that nitration and oxidation of proteins involved in self-recognition results in sustained inflammatory response mediated by neutrophils and macrophages. Activation of these immune cells results in increased NADPH oxidase synthesis which in turn perpetuates peroxynitrite-modification of proteins [13].”
Reviewer 2 Report
This brief review from Sylvester et al briefly summarizes some details of ongoing clinical trials using Nox inhibitors to reduce tissue damage in number of different diseases. More specific detail and descriptions about the pathways and disease related pathologies will improve the quality of this review. Also, the distinct focus on vascular function is unclear. I have listed some suggestions below that I hope the authors will find helpful.
Major:
- The title is misleading, I would suggest altering. There is no explanation given in the text for how microvascular function is involved in PBC and IPF. Hepatocytes and respiratory epithelial cells have NOX isoforms and there is no indication or explanation as to why these trials are related to vascular NOX inhibition specifically.
- The Introduction would benefit from a bit more organization. From an initial read, it seems as though the review will be discussing the clinical trials, but it segues into a brief historical summary before returning to the discussion of clinical trials. I would suggest rearranging things so a brief overview of ROS and NOXs is given, then put into historical context of how they were discovered, then describe in more detail their effects in disease (as well as how in some cases they can be beneficial), and then go on to describe the current/upcoming clinical trials by which NOX inhibition will improve patient outcomes.
- The first three introductory paragraphs are quite broad and could benefit from inclusion of some more specific details regarding what is being defined as a pathology, i.e. give examples of specific diseases and their related damage.
- Was there a reason to exclude the Type 2 Diabetes and Albuminuria clinical trial by Genkyotex?
- I do not understand what “Despite ineffectiveness of efforts aiming at full elimination of ROS” means. Is this from KO animal studies?
Minor:
- Define ROS and CAD in the text the first time they are used.
- Reference 1 is from 2013, but sets up an important concept about the inconclusive evidence of antioxidant supplementations. Are there any more recent and disease-specific publications relating to the diseases discussed in the clinical trials that could also be included here? This reference is also quite broad referring only to “pathologies.” What kind of pathologies?
Author Response
Dear Reviewer:
Thank you for the thoughtful comments and careful review of our manuscript. We have thoroughly addressed each of the comments in the revised manuscript. Please see our responses below.
This brief review from Sylvester et al briefly summarizes some details of ongoing clinical trials using Nox inhibitors to reduce tissue damage in number of different diseases. More specific detail and descriptions about the pathways and disease related pathologies will improve the quality of this review. Also, the distinct focus on vascular function is unclear. I have listed some suggestions below that I hope the authors will find helpful.
Major:
- The title is misleading, I would suggest altering. There is no explanation given in the text for how microvascular function is involved in PBC and IPF. Hepatocytes and respiratory epithelial cells have NOX isoforms and there is no indication or explanation as to why these trials are related to vascular NOX inhibition specifically.
Thank you for your kind suggestion. The title was altered to include other pathologies in the scope of this review. Additionally, expression of NOX isoforms in hepatocytes and respiratory epithelial cells and specific roles of individual isoforms in disease and pathogenesis is emphasized to highlight why these clinical trials are related to vascular NOX inhibition.
“NADPH oxidases have been implicated in numerous pathologies and pathology-driving pathways. Pulmonary epithelial cells express NOX isoforms including NOX1, NOX2, and NOX4 [37]. IPF is characterized by increased levels of mitochondrial and NADPH oxidase ROS. The condition is exacerbated by dysfunctional mitochondria which produce excess O2- and H2O2, increasing expression of NOX4 and TGFβ-1 signaling [2][38]. TGFβ-1 signaling promotes proinflammatory damage and collagen accumulation in the lungs [39], which in turn drives apoptosis and the formation of fibrotic tissue in IPF [40]. NOX4 additionally mediates the activity of TGFβ-1-induced cell differentiation, cardiac differentiation and transcriptional regulation [19]. Hepatocytes also express NOX isoforms, including NOX1, NOX2 and NOX4, which have been implicated in critical steps in initiating liver fibrosis, including hepatic stellate cell activation and hepatocyte apoptosis [3]. TGFβ-1 has also been demonstrated to contribute to the progression of PBC by enhancing fibrogenesis [41]. Together, these data suggest NADPH oxidases, particularly NOX4, may play a role in the progression of fibrotic pathologies such as IPF and PBC, implicating selective NADPH oxidase inhibitors as promising therapeutic agents.”
- The Introduction would benefit from a bit more organization. From an initial read, it seems as though the review will be discussing the clinical trials, but it segues into a brief historical summary before returning to the discussion of clinical trials. I would suggest rearranging things so a brief overview of ROS and NOXs is given, then put into historical context of how they were discovered, then describe in more detail their effects in disease (as well as how in some cases they can be beneficial), and then go on to describe the current/upcoming clinical trials by which NOX inhibition will improve patient outcomes.
The introduction has been revised to address these suggestions. We restructured the introduction by beginning with a brief overview of ROS and NOXs including their definitions followed by a brief historical summary. We then described the past failure of clinical trials using nonspecific antioxidants as well as experimental evidence implicating specific NOX isoforms in disease and underlying pathways. This is followed by description of recent clinical trials that employ specific NOX inhibitors including the major goals of these trials and the reasons for their potential success.
- The first three introductory paragraphs are quite broad and could benefit from inclusion of some more specific details regarding what is being defined as a pathology, i.e. give examples of specific diseases and their related damage.
The first three introductory paragraphs have been extensively revised. The initial information included in these paragraphs (brief history of NADPH oxidases and ROS) have been moved to the Introduction section. The revised paragraphs contain more specific information describing: 1) NOX isoforms expression pattern in various tissues; 2) the proposed roles for NOX isoforms in pathologies; 3) recent experimental studies exploring the roles of ROS and NOX in health and disease; and 4) specific diseases driven by NADPH oxidases through various mechanisms. This information is directly related to description of the clinical trials in patients with diabetes, IPF, and PBC.
- Was there a reason to exclude the Type 2 Diabetes and Albuminuria clinical trial by Genkyotex?
Thank you for this suggestion. The revised review has been updated to include the type 2 Diabetes and Albuminuria clinical trial by Genkyotex; the table displaying clinical trials has also been updated.
“Type 2 diabetes is a progessive disease in which hyperglycemia is strongly associated with stroke, myocardial infarction, and mortality. Hallmarks include loss of β-cell function and decrease in insulin resistance [59]. Diabetes is the leading cause of chronic kidney disease, frequently resulting in albuminuria [60]. Renin-angiotensin-aldosterone inhibition through either an ACE inhibitor or angiotensin receptor antagonist is the current treatment guideline for albuminuria, and is reported to slow the progression of kidney disease [61]. Oxidative stress due to ROS may initiate the progression of vascular and endothelial dysfunction associated with type 2 diabetes [23]. Thus, a clinical trial evaluating the efficacy of oral GKT137831, a NOX1/NOX4 inhibitor, in 200 type 2 diabetes patients with maximal inhibition of the renin-angiotensin-aldosterone system and residual albuminuria was conducted. The study began in October 2013 and concluded in March 2015. The primary outcome measures of the study were albuminuria and urine albumin-to-creatinine ratio (ACR), a useful predictor for cardiovascular outcomes and mortality in patients with diabetes [62]. Secondary outcome measures were glucose metabolism and changes in HOMA-B, HOMA-IR, and HbA1c from baseline. Additional outcome measures included erectile dysfunction and neuropathic pain in patients with these complications. Two study arms adhered to either of the following regimens: 1 100mg capsule twice per day for 6 weeks followed by 2 100mg capsules twice per day for 6 weeks, or 1 capsule of placebo twice per day for 12 weeks [63]. While this study has been completed, the results were not available to include in this publication. The authors of this study have been contacted and the results will be updated.”
- I do not understand what “Despite ineffectiveness of efforts aiming at full elimination of ROS” means. Is this from KO animal studies?
We apologize for this inaccurate statement that was now deleted. The revised statement emphasizes the failure of clinical trials using nonspecific antioxidants without commenting on full elimination of ROS or KO animals. This sentence has been edited to:
“However, clinical trials using nonspecific antioxidants have consistently failed to demonstrate measurable benefits.”
Minor:
- Define ROS and CAD in the text the first time they are used.
Thank you for this reminder. All acronyms are now defined upon first mentioning in the text.
“Release of reactive oxygen species (ROS), which are unstable oxygen-containing molecules that easily react with other molecules, is central to the maintenance of vascular homeostasis.”
“This is transient in healthy subjects but permanent in patients with coronary artery disease (CAD), a condition in which coronary arteries send insufficient blood to the heart, typically due to atherosclerosis and inflammation.”
- Reference 1 is from 2013, but sets up an important concept about the inconclusive evidence of antioxidant supplementations. Are there any more recent and disease-specific publications relating to the diseases discussed in the clinical trials that could also be included here? This reference is also quite broad referring only to “pathologies.” What kind of pathologies?
The text has been updated to include more recent and disease-specific publications relevant to the clinical trials described in this review. Updated references refer to CAD and all-cause mortality. We refer to these specific causes that replace the broad reference to “pathologies”.
“A 2020 meta-analysis of 43 antioxidant clinical trials show lack of association between all-cause mortality and cardiovascular disease (CVD) and antioxidant supplementation alone. A 2022 study shows that association between CVD risk and vitamins A and C, and zinc was not statistically significant though vitamin E may reduce CVD risk.”
Reviewer 3 Report
The review by Anthony L. Sylvester et al, provides an overview of the important role of NADPH oxidases enzymes in human vascular function. Overall the topic could be interesting but some details could be improved.
I recommend that the review be accepted with major revision:
- In the title the authors talk about of experimental studies, but they are missing in the text. they should add at least one paragraph on the role of ROS and NADPH oxidase in experimental studies. Please referee doi: 10.3389/fcell.2021.628991; 10.2174/0929867328666210329120213; 10.1089/ars.2013.5607.
- The authors should better emphasize the conclusions.
- The authors should briefly describe the NADPH oxidases.
- The authors should provide an illustrated figure to summarize and simplify about NADPH oxidases enzymes in human vascular.
- The literature is poorly exhaustive and updated. Please add recent references.
Author Response
Dear Reviewer:
Thank you for the thoughtful comments and careful review of our manuscript. We have thoroughly addressed each of the comments in the revised manuscript. Please see our responses below.
The review by Anthony L. Sylvester et al., provides an overview of the important role of NADPH oxidases enzymes in human vascular function. Overall the topic could be interesting but some details could be improved.
I recommend that the review be accepted with major revision:
- In the title the authors talk about of experimental studies, but they are missing in the text. they should add at least one paragraph on the role of ROS and NADPH oxidase in experimental studies. Please referee doi: 10.3389/fcell.2021.628991; 10.2174/0929867328666210329120213; 10.1089/ars.2013.5607.
Thank you for this excellent suggestion. Section 2.2 now describes recent experimental studies addressing the role of ROS and NADPH oxidase in health and disease. The second listed publication is also included in this section. Since the first suggested paper is a review, experimental studies described within are included in cited references; our manuscript then refers to these citations to explain conclusions drawn from these studies. We did not include the third recommended paper from 2014 in this section as it did not contain studies directly related to the topic of the section. However, it was referenced in the subsequent section 2.3 to describe pathological pathways such as the AT-II/NOX2 that contribute to atherosclerosis.
- The authors should better emphasize the conclusions.
We added significantly more details on experimental studies, mechanisms of ROS generation, their impact on development of pathologies in human diseases, and possible solutions for counteracting their detrimental effects. We believe that the conclusions in the revised manuscript are now based on much broader evidence-based information, and therefore, justified.
- The authors should briefly describe the NADPH oxidases.
NOX isoforms tissue expression patterns and related pathologies have been now included in section 2.1.
“Four NADPH oxidases (NOX1, NOX2, NOX4, and NOX5) are relevant to vascular homeostasis and pathology. NOX1 is widely distributed in various cell types, but expression is particularly abundant in colonic epithelium and vascular smooth muscle cells [20]. It has been implicated in colon cancer progression [17] and vascular complications in diabetes [1]. The major ROS source in humans is NOX2 [21] that is highly expressed in phagocytes [22]. It is also the most widely expressed NOX isoform [19]. NOX2 contributes to endothelial dysfunction in vascular pathologies such as insulin resistance in diabetes [23], but may also mediate phenotypic conversion of macrophages for tissue repair [24]. NOX4 is abundant in non-phagocytic cells, and it has been detected in vascular walls, fibroblasts, endothelial cells, and the kidney [19]. It mediates proinflammatory TGFβ-1 signaling in diseases such as IPF [19], but also is necessary for polarization of macrophages [25]. NOX5 expression has been detected in lymphatic tissue, the testis, and blood vessels in humans [26]. Some studies showed that NOX5 contributes to vascular and kidney pathologies [26], but a recent study demonstrated a potential protective role of NOX5 against atherosclerosis in rabbits [27].”
- The authors should provide an illustrated figure to summarize and simplify about NADPH oxidases enzymes in human vascular.
Thank you for the critique. We have included figure 1 that summarizes and simplifies the role of NADPH oxidase and other ROS-producing enzymes in human vasculature to address this issue.
- The literature is poorly exhaustive and updated. Please add recent references.
The manuscript has been updated to include more specific information drawn from multiple recent studies. Additionally, the number of references has been expanded to reflect numerous published studies regarding NADPH oxidases and ROS.
Round 2
Reviewer 1 Report
The manuscript is improved and can be accepted.
Reviewer 2 Report
Nice work with the updated information, I have nothing else to suggest.
Reviewer 3 Report
The authors adequately answered the reviewers' questions.
I recommend that the paper be accepted in current form.